# Estimating the cumulative incidence of COVID-19 in the United States using influenza surveillance, virologic testing, and mortality data: Four complementary approaches

Fred S. Lu[1], Andre T. Nguyen[2,3], Nicholas B. Link[4], Mathieu Molina[4], Jessica T. Davis[5], Matteo Chinazzi[5], Xinyue Xiong[5], Alessandro Vespignani[5], Marc Lipsitch[6], Mauricio Santillana[4,6,7] *

1 Department of Statistics, Stanford University, Stanford, California, United States of America, 2 University of Maryland, Baltimore County, Baltimore, Maryland, United States of America, 3 Booz Allen Hamilton, Columbia, Maryland, United States of America, 4 Computational Health Informatics Program, Boston Children's Hospital, Boston, Massachusetts, United States of America, 5 Laboratory for the Modeling of Biological and Socio-technical Systems, Northeastern University, Boston, Massachusetts, United States of America, 6 Department of Epidemiology, Harvard T.H. Chan School of Public Health, Boston, Massachusetts, United States of America, 7 Department of Pediatrics, Harvard Medical School, Boston, Massachusetts, United States of America

☯ These authors contributed equally to this work.
* msantill@fas.harvard.edu

**Data Availability Statement:** The data is held in a public repository at https://github.com/Machine-Intelligence-Group/covid-incidence-estimation. The

## Abstract

Effectively designing and evaluating public health responses to the ongoing COVID-19 pandemic requires accurate estimation of the prevalence of COVID-19 across the United States (US). Equipment shortages and varying testing capabilities have however hindered the usefulness of the official reported positive COVID-19 case counts. We introduce four complementary approaches to estimate the cumulative incidence of symptomatic COVID-19 in each state in the US as well as Puerto Rico and the District of Columbia, using a combination of excess influenza-like illness reports, COVID-19 test statistics, COVID-19 mortality reports, and a spatially structured epidemic model. Instead of relying on the estimate from a single data source or method that may be biased, we provide multiple estimates, each relying on different assumptions and data sources. Across our four approaches emerges the consistent conclusion that on April 4, 2020, the estimated case count was 5 to 50 times higher than the official positive test counts across the different states. Nationally, our estimates of COVID-19 symptomatic cases as of April 4 have a likely range of 2.3 to 4.8 million, with possibly as many as 7.6 million cases, up to 25 times greater than the cumulative confirmed cases of about 311,000. Extending our methods to May 16, 2020, we estimate that cumulative symptomatic incidence ranges from 4.9 to 10.1 million, as opposed to 1.5 million positive test counts. The proposed combination of approaches may prove useful in assessing the burden of COVID-19 during resurgences in the US and other countries with comparable surveillance systems.

GLEAM model is publicly available at www.gleamviz.org/. Epidemic surveillance data used to calibrate GLEAM were collected from the Johns Hopkins Coronavirus Resource Center https://coronavirus.jhu.edu/. Proprietary airline data are commercially available from OAG (https://www.oag.com/) and IATA (https://www.iata.org/) databases. Intervention data used in GLEAM includes Google's COVID-19 Community Mobility Reports available at https://www.google.com/covid19/mobility/ and the Oxford COVID-19 Response Tracker available at https://github.com/OxCGRT/covid-policy-tracker.

**Funding:** MS and AV are partially supported by the National Institute of General Medical Sciences of the National Institutes of Health under Award Number R01GM130668. The content is solely the responsibility of the authors and does not necessarily represent the official views of the National Institutes of Health. The funders had no role in study design, data collection and analysis, decision to publish, or preparation of the manuscript.

**Competing interests:** I have read the journal's policy and the authors of this manuscript have the following competing interests: ML has provided advice on COVID-19 free of charge to Janssen, Astra-Zeneca, Pfizer, and COVAXX (United Biomedical), as well as to the nonprofit One Day Sooner. ML has received consulting income or honoraria from Merck, Pfizer, Bristol Meyers Squibb, and Sanofi, and institutional research support from Pfizer.

## Author summary

Accurate estimates of the weekly incidence of COVID-19 in the United States is essential for planning and researching effective public health responses. Because of systematic testing shortages across the United States, official positive COVID-19 test counts are an unreliable indicator of true incidence. In this study, we present four alternative approaches for estimating cumulative incidence, which leverage different data sources and assumptions. Nationally, our estimates of COVID-19 symptomatic cases as of April 4 have a likely range of 2.3 to 4.8 million, with possibly as many as 7.6 million cases, up to 25 times greater than the cumulative confirmed cases of about 311,000. We emphasize that comparing multiple models rather than relying on a single method gives more reliable estimates of COVID-19 incidence. Our approaches could be useful for tracking the resurgence of COVID-19 in the United States as well as in other countries.

## Introduction

COVID-19 (SARS-CoV-2), is a coronavirus that was first identified in Hubei, China, in December of 2019. On March 11, due to its extensive spread, the World Health Organization (WHO) declared it a pandemic [1]. As of July 24, 2020, COVID-19 had infected people in nearly every country globally with an official case count surpassing 15 million cases worldwide and 4 million in the United States (US) [2]. It is however accepted that the official case count is capturing only a fraction of the actual infections, and reliable estimates of COVID-19 infections are critical for appropriate resource allocation, effective public health responses, and improved forecasting of disease burden [3].

A lack of widespread testing due to equipment shortages, varying levels of testing by region over time, and uncertainty around test sensitivity make estimating the point prevalence of COVID-19 difficult [4, 5]. In addition, meta-analyses have estimated that 17% [6] or 20% [7] to 45% [8] of people infected with COVID-19 are asymptomatic or paucisymptomatic. Even in symptomatic infections, under-reporting can further complicate the accurate characterization of the COVID-19 burden. For example, one study estimated that in China, 86% of cases had not been captured by lab-confirmed tests [9], and it is possible that this percentage is even higher in the US [5]. Finally, it has been suggested that the available information on confirmed COVID-19 cases across geographies may be an indicator of the local testing capacity over time, as opposed to an indicator of the epidemic trajectory. Thus, solely relying on positive test counts to infer the COVID-19 epidemic trajectory may not be sensible [10].

The aim of this study is to show how alternative methodologies, each with different sets of inputs and assumptions, can provide a consensus estimate of weekly cumulative symptomatic incidence of COVID-19 in each state in the US. One such approach is to analyze region-specific changes in the number of individuals seeking medical attention with influenza-like illness (ILI), defined as having a fever in addition to a cough or sore throat. The significant overlap in symptoms common to both ILI and COVID-19 suggests that leveraging existing disease monitoring systems, such as ILINet, a sentinel system created and maintained by the United States Centers of Disease Control and Prevention (CDC) [11, 12], may offer a way to estimate the ILI-symptomatic incidence of COVID-19 without needing to rely on COVID-19 testing results. Importantly, regional increases in ILI observed from February to April 2020 in conjunction with stable or decreasing influenza case numbers present a discrepancy (i.e., an increase in ILI not explained by an increase in influenza) that can be used to impute COVID-19 ILI-symptomatic cases. We denote such methods as the *Divergence* approach.

A second and related approach (denoted as *COVID Scaling*) uses ILI data to deconfound COVID-19 testing results from state-level testing capabilities. These two approaches show that existing ILI surveillance systems are a useful signal for measuring COVID-19 ILI-symptomatic incidence in the US, especially during the early stages of the outbreak. However, they are dependent on reporting from the ILINet system, and thus become less reliable outside of peak flu season and when COVID-19 precautions disrupt routine health care use.

Our third approach (denoted as *mMAP*) uses reported COVID-19-attributed deaths to estimate COVID-19 symptomatic incidence (broader than the ILI-symptomatic incidence of the first two methods) and improves upon previously introduced methodologies [13–17]. COVID-19 deaths may represent a lower-noise estimate of cases than surveillance testing given that patients who have died are sicker, more likely to be hospitalized, and thus more likely to be tested than the general infected population.

The fourth approach is based on the use of the Global Epidemic and Mobility model (*GLEAM*), a fully stochastic epidemic modeling platform that uses real-world data to perform *in silico* simulations of the spatial spread of COVID-19 in the US [18]. The mechanistic modeling stage explores the parameter space defined by the basic reproduction number, generation time, seasonality scaling factor, social distancing policies, and generates a corpus of simulated epidemic profiles. The simulation results can be aggregated at the level of each US state and the entire country. The model selection stage is performed by measuring the information loss with respect to the ground truth surveillance data of the weekly death incidence in each state.

While previous work has attempted to quantify COVID-19 incidence in the United States using discrepancies in ILI trends [19, 20], to the best of our knowledge this study is the first to offer a range of estimates at the state level, leveraging a suite of complementary methods based on different assumptions. We believe that this provides a more balanced picture of the uncertainty over COVID-19 (ILI-)symptomatic incidence in each state. While our results are approximations and depend on a variety of (likely time-dependent) estimated factors, we believe that our presented case counts better represent (ILI-)symptomatic incidence than simply relying on laboratory-confirmed COVID-19 tests. Providing such estimates for each state enables the design and implementation of more effective and efficient public health measures to mitigate the effects of the ongoing COVID-19 epidemic outbreak. While the scope of this paper is focused on the United States, the methods introduced here are general enough that they may prove useful to estimate COVID-19 burden in other locations with comparable disease (and death) monitoring systems.

## Results

We implement four approaches—*Divergence*, *COVID Scaling*, *mMAP*, and *GLEAM*—to estimate the cumulative symptomatic incidence of COVID-19 within the US from March 1 to April 4, 2020 (we further extend *mMAP* and *GLEAM* predictions to May 16, 2020). These dates correspond to the early stages of the outbreak (with fewer than 50 confirmed cases in the US), up to the date of the the CDC reports as of May 28th, 2020. Two methods, labeled *div-Hist* and *div-Vir*, fall under the *Divergence* approach, which first estimates what the level of ILI activity across the US would have been if the COVID-19 outbreak had not occurred. Each method develops a control time series and uses the unexpected increase in the ILI rate over the control to infer the burden of COVID-19. *div-Hist* is based on a seasonal time series decomposition, fitted to the observed 2019–2020 ILI (prior to the introduction of COVID-19 to the US), while *div-Vir* is based on the time-evolution of empirical observations of positive virological influenza test statistics. A third method, using the *COVID Scaling* approach, leverages healthcare ILI visits and COVID-19 test statistics to directly infer the proportion of ILI due to

COVID-19 in the full population. These three methods estimate ILI symptomatic incidence and may miss symptomatic patients not matching the ILI symptoms (for the remainder of the paper, we use 'ILI-symptomatic' to denote COVID-19 patients with ILI symptoms and 'symptomatic' to denote COVID-19 patients with any symptoms). In addition, these methods are accurate only while ILI surveillance systems are operating normally (usually only during the flu season) and only while the outbreak has not yet overwhelmed hospitals. We use the ILI based methods to estimate ILI-symptomatic case counts until April 4th, 2020.

The fourth method, using the *mortality MAP* (*mMAP*) approach, uses the time series of reported COVID-19-attributed deaths in combination with the observed epidemiological characteristics of COVID-19 in hospitalized individuals to infer the latent disease onset time series. This is then scaled up to yield estimates of symptomatic case counts using reported estimates of the symptomatic case fatality rate (sCFR). Finally we use a fifth method based on the explicit modeling of the epidemic using the *GLEAM* model, calibrated on reported deaths. The model provides the number of individuals that have been infected, the number of individuals that are currently infectious, and the number of daily new infections in US states and at the national level. *GLEAM* estimates the cumulative number of both symptomatic and asymptomatic infections using an estimated infection fatality rate (IFR) [21], so it is scaled down by 40%, the current best point estimate for the number of infections that are asymptomatic [8, 22, 23], to produce estimates of symptomatic cases. The Methods section provides extensive details on the assumptions and data sources for each of these approaches.

## Adjusted assumptions represent most likely scenarios

Each method from the first three approaches has an adjusted version, which represents our best guess taking into account all information available to us, and an unadjusted version, which uses pre-COVID-19 baseline information. Specifically, the adjusted divergences (*div-Hist* and *div-Vir*) and *COVID Scaling* methods incorporate an increased probability that an individual with ILI symptoms will seek medical attention after the start of the COVID-19 outbreak based on recent survey data [24, 25]. The adjusted *mMAP* incorporates newer information from serological testing, indicating a lower IFR and asymptomatic rate (and thus higher estimated symptomatic case count) than expected. In addition, it supplements the confirmed COVID-19 deaths with unusual increases in influenza and pneumonia-related deaths across the country that may represent untested COVID-19 cases. Since there is no unadjusted version for *GLEAM* and because its sCFR (calculated as $\frac{IFR}{1-AR}$, $AR$ = asymptomatic rate) is the same as the sCFR in adjusted *mMAP*, we group *GLEAM* in with the adjusted methods. In most states, as seen in Fig 1, the adjusted estimates from each method are more closely clustered than their unadjusted counterparts, increasing our confidence in the adjusted range estimates of COVID-19 cumulative symptomatic incidence (ILI-symptomatic specifically for the ILI based methods).

## Estimated case counts far surpass reported positive cases

We first computed estimates for the national and state levels (including the District of Columbia and Puerto Rico) using these four approaches for the time period between March 1, 2020 and April 4, 2020. The adjusted methods estimate that there had been 2.3 to 4.8 million symptomatic or ILI-symptomatic COVID-19 cases in the US; including unadjusted estimates raises the upper limit to 7.6 million cases. In comparison, around 311,000 positive cases had been officially recorded during that time period. Fig 1A displays the COVID-19 symptomatic case count estimates from our methods (ILI-symptomatic in particular for the ILI based methods) at the national and state levels compared with the reported case numbers. The results suggest

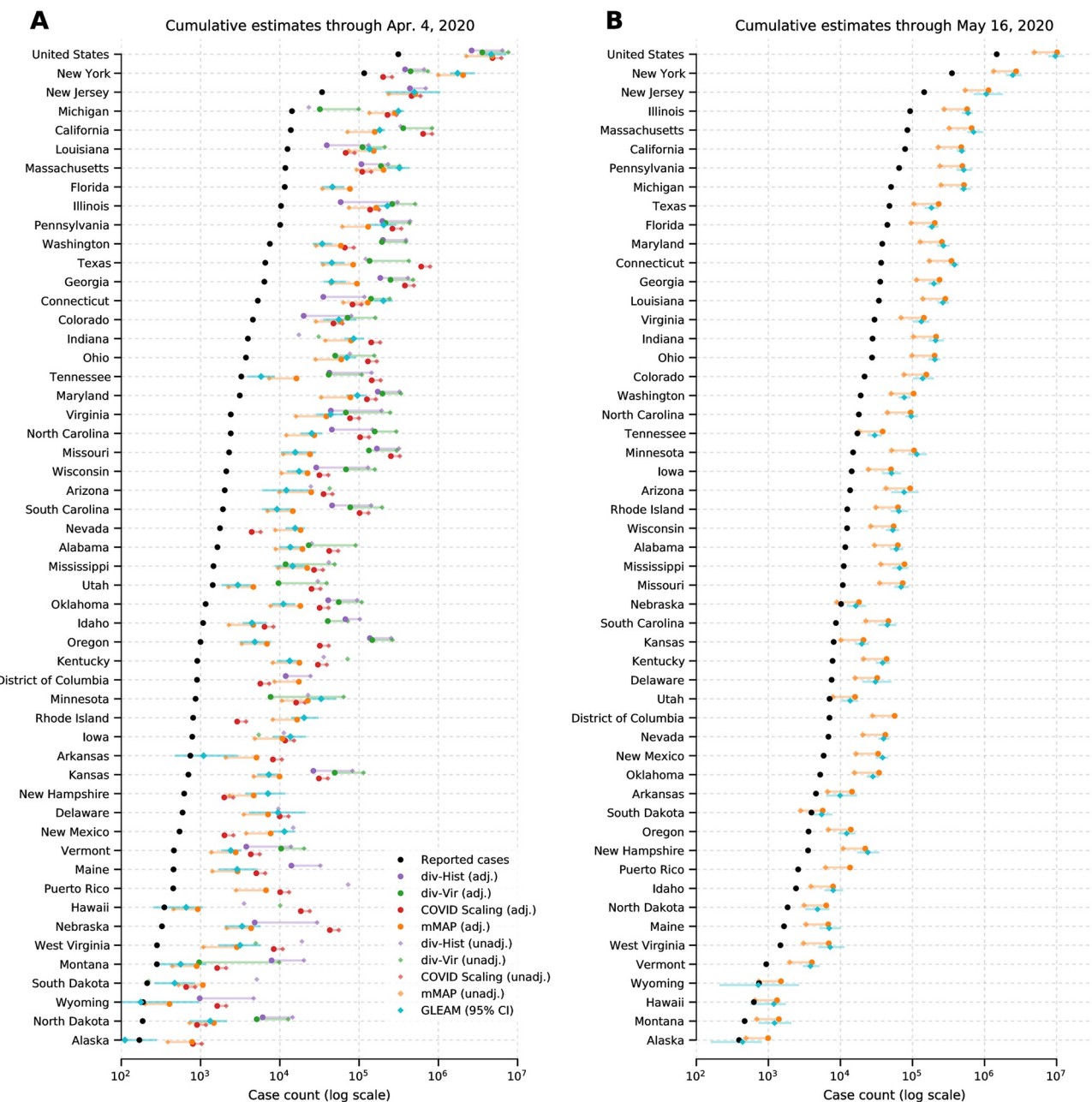

**Fig 1. COVID-19 (ILI-)symptomatic case count estimates compared with reported case counts at the national and state levels from March 1, 2020 to (A) April 4, 2020 and (B) May 16, 2020.** Cases are presented on a log scale. Adjusted methods take into account increased visit propensity (*div-Hist*, *div-Vir*, *COVID Scaling*) and excess influenza and pneumonia deaths along with a lower estimated case fatality rate (*mMAP*). In places where the ILI-based methods show no divergence in observed and predicted ILI visits, the estimates of COVID-19 cannot be calculated and are not shown. Note that Florida does not provide ILI data, so only *mMAP* could be estimated there.

that the estimated true numbers of infected cases are nearly uniformly much higher than those reported. Next, we extended our methods to produce estimates through May 16, 2020 using recent data, displayed in Fig 1B. Because of a strong decline in ILINet statistics due to the end of the flu season and unusually low numbers of reporting providers, our *Divergence* and *COVID Scaling* approaches report few or no cases after April 4, 2020. Therefore, our recent

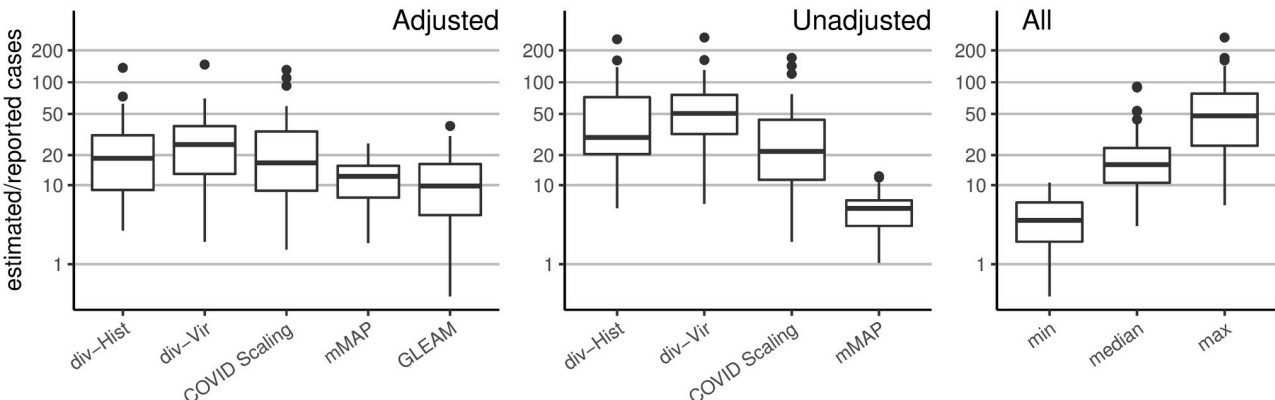

**Fig 2. Distribution of the state-level ratios of estimated to reported case counts from March 1, 2020 to April 4, 2020.** The right-hand plot shows the results of using all methods together: taking the *min*, *median*, and *max* of the state-level estimates across methods. Adjusted methods take into account increased visit propensity (*div-Hist*, *div-Vir*, *COVID Scaling*) and excess influenza and pneumonia deaths along with a lower estimated case fatality rate (*mMAP*).

estimates are computed using the *mMAP* method and the *GLEAM* model, which estimate between 4.9 and 10.1 million symptomatic cases had occurred as of May 16. In contrast, 1.5 million positive test counts had been reported. This highlights that models using only confirmed test cases may significantly underestimate the actual COVID-19 cumulative incidence in the United States, which is consistent with what previous studies have shown [9, 20].

As a naive baseline, if one only adjusts the number of reported cases by the (likely) percentage of asymptomatic cases (18% [6, 26] to 50% [27, 28]) and symptomatic cases not seeking medical attention (up to 73% [29]), one would conclude that the actual number of cases were about four to eight times the number of reported cases; this ratio would also be constant across states. In contrast, our methods frequently estimate 5-fold to 50-fold more symptomatic (for *mMAP*) or ILI-symptomatic (for *Divergence* and *COVID Scaling*) cases than those reported and show significant state-level variability (see Fig 2). The median estimates for the ratios of estimated cases to reported cases from March 1 to April 4, 2020 for the adjusted *div-Hist* method is 18 (with a 90% interval from 1 to 101), for adjusted *div-Vir* is 21 (2, 67), for adjusted *COVID-Scaling* is 17 (3, 76), for adjusted *mMAP* is 11 (4, 20), and for *GLEAM* is 10 (2, 29).

Using our methods, we also compute cumulative case estimates for each week within the studied period. Fig 3 highlights the rapid increase in estimated COVID-19 cases over the United States as well as in New York, Washington, and Louisiana, three locations which experienced early outbreaks. These methods suggest that states under-reported COVID-19 case counts even early in March, likely due to limited testing availability. In New York and Louisiana, the estimates were more similar across methods than in Washington. Since Washington had already experienced an outbreak by February 28 [30], testing shortages may have been more pronounced than in the other states. Our divergence analysis approach does not rely on any COVID-19 test-dependent data (including deaths) and therefore may provide more accurate estimates in Washington.

## State-level comparisons

Over the period of March 1, 2020 to April 4, 2020, the adjusted *div-Hist*, *div-Vir*, *COVID Scaling*, *mMAP*, and *GLEAM* approaches estimated that between 21 and 35 (21, 25, 35, 35, and 25, respectively) locations had actual (ILI-)symptomatic case counts above 10 times the reported counts (Figs 1 and 2). Up to 12 locations had at least one adjusted estimate above 50 times the

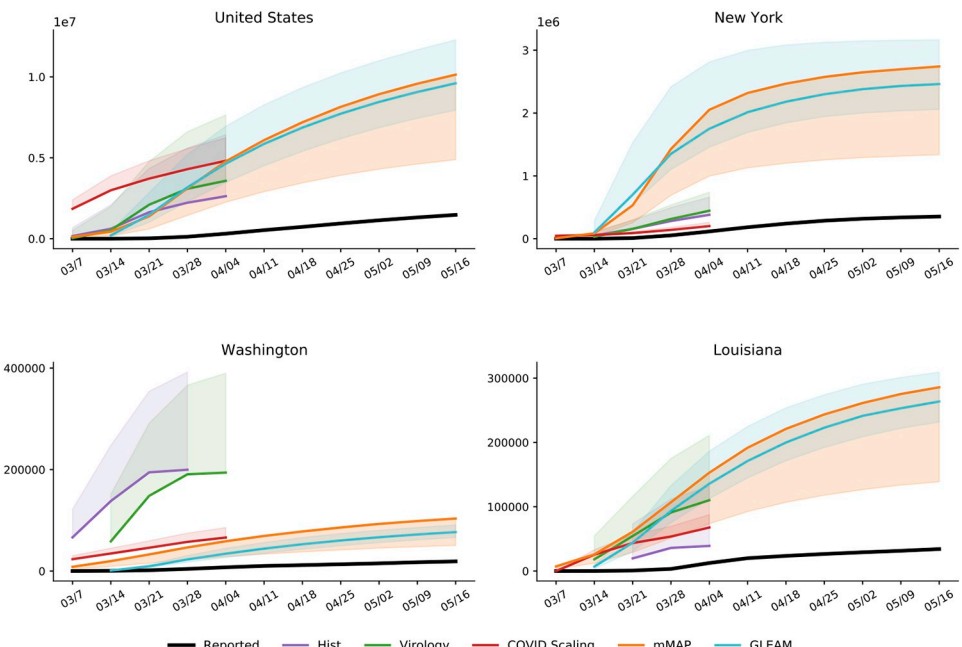

**Fig 3. Cumulative weekly case counts from March 1 to May 16, 2020 for the United States, New York, Washington, and Louisiana, as estimated by each method and the reported cases.** The estimate for each week indicates total cases up to the denoted date. Solid lines indicate the adjusted estimates with shading for the unadjusted estimate ranges. Adjusted methods take into account increased visit propensity (*div-Hist*, *div-Vir*, *COVID Scaling*) and excess influenza and pneumonia deaths along with a lower estimated case fatality rate (*mMAP*). Refer to S2 Fig for results over all locations.

reported counts, with three of them above 100 times the reported counts (Nebraska, Oregon, Missouri). Places with low official case counts, such as Alaska and North Dakota, may have experienced significantly more COVID-19 cases than reported. Even places with high official case counts, such as Georgia, Pennsylvania, and Texas, appeared to be significantly under-reporting. As expected, our methods computed high estimates in New York and New Jersey, locations with especially high numbers of confirmed cases. Over the period leading up to May 16, 2020, *mMAP* and *GLEAM* estimates indicate that up to 30 locations had estimated case counts above five times the reported counts, with two locations over 10 times (Connecticut and Michigan).

Using the unadjusted methods, the ILI-based methods yield significantly higher estimates than *mMAP* (median estimates of 84k, 155k, 62k, 11k for *div-Hist*, *div-Vir*, *COVID Scaling*, and *mMAP*, respectively, for the locations that have estimates for all methods). However, the adjusted versions of the methods (including *GLEAM*) are more similar (median estimates of 35k, 73k, 85k, 23k, and 17k for *div-Hist*, *div-Vir*, *COVID Scaling*, *mMAP*, and *GLEAM*), providing support that the adjusted methods are more accurate than the unadjusted ones.

All five methods generally agree on the ordering of states by (ILI-)symptomatic case count (Table 1), with rank correlations of the adjusted methods ranging from 0.64 to 0.98. *mMAP* and *GLEAM* have 0.95 and 0.91 correlations with the reported case counts, which is likely because official COVID-19 deaths and positive COVID-19 cases represent overlapping pools of patients and are therefore subject to similar biases. *COVID Scaling* also shows a relatively high correlation with the reported cases, 0.88, which may reflect the use of COVID-19 test statistics in its model. *div-Hist* and *div-Vir*, however, solely rely on aggregate data from ILINet, which may cover a different set of patients.

**Table 1. Pairwise Spearman correlations between adjusted methods and reported case counts from March 1, 2020 to April 4, 2020 across the state level.**

|  | Reported | div-Hist | div-Virology | COVID Scaling | mMAP | GLEAM |
|---|---|---|---|---|---|---|
| Reported | 1.00 | 0.70 | 0.71 | 0.88 | 0.95 | 0.91 |
| div-Hist | – | 1.00 | 0.78 | 0.70 | 0.67 | 0.64 |
| div-Virology | – | – | 1.00 | 0.69 | 0.68 | 0.66 |
| COVID Scaling | – | – | – | 1.00 | 0.84 | 0.78 |
| mMAP | – | – | – | – | 1.00 | 0.98 |
| GLEAM | – | – | – | – | – | 1.00 |

## Discussion

We present five methods based on four distinct approaches to estimate the COVID-19 cumulative symptomatic incidence across the United States. The methods are complementary, in that they rely on different methods, assumptions and use diverse datasets. Despite their clear differences, these methods estimate that the likely COVID-19 cumulative symptomatic incidence varies from 5 to 50 times higher, at the state level, than what has been reported so far in the U.S. By providing ranges of estimates, both within and across models, our suite of methods offers a robust picture of the under-ascertainment of state-level COVID-19 case counts. When making public health decisions to respond to COVID-19, it is important to account for the uncertainty in estimates of symptomatic incidence; the multiple estimates presented here provide a consistent picture of the number of infected individuals.

Our estimates are specifically for symptomatic cases, while a high proportion of COVID-19 cases are believed to be asymptomatic [23, 26, 27]. To estimate total cases, our counts can be adjusted by the proportion of symptomatic cases. For example, if 40% of cases are asymptomatic, this could indicate a total cumulative incidence of up to 16.8 million as of May 16, 2020.

Our approaches could be expanded to include other data sources and methods to estimate incidence, such as Google searches [31–33], electronic health record data [34], clinician's searches [35], and/or mobile health data [36]. Accurate and appropriately sampled serological testing would provide the most accurate estimate of incidence and would be useful for public health measures, especially when attempting to relax or re-institute shelter-in-place recommendations. In addition, serological testing could be used to evaluate the reliability of the methods presented in this study. This could inform prevalence estimation methods for COVID-19 in other countries as well as for future pandemics. The ILI-based methods presented in this study demonstrate the potential of existing and well-established ILI surveillance systems to monitor future pandemics that, like COVID-19, present similar symptoms to ILI. This is especially promising given the WHO initiative launched in 2019 to expand influenza surveillance globally [37]. Incorporating estimates from influenza and COVID-19 forecasting and participatory surveillance systems may prove useful in future studies as well [18, 38–42].

### Limitations

Since the *Divergence* and *COVID Scaling* approaches are estimated using ILINet statistics, their symptomatic incidence estimates are dependent on the ILI definition of a fever and cough or sore throat. Thus, they may miss a percentage of COVID-19 patients that are symptomatic without meeting the ILI definition. With this limitation, the reported estimates may serve as an approximate lower bound. Given a clearer understanding of COVID-19 symptoms, our *Divergence* and *COVID Scaling* estimates could be adjusted upward by the proportion of symptomatic to ILI-symptomatic patients.

Furthermore, we note that ILI surveillance networks may not always accurately measure ILI for the most at risk elderly individuals who reside in nursing homes.

As well, the data used and delays in reporting affect the timing of the methods' estimates. The *Divergence* and *COVID Scaling* methods estimate the date of medical visitation for ILI/COVID-19 symptoms while the *mMAP* and *GLEAM* methods estimate the date of COVID-19 symptom onset, which is expected to be on average 4–5 days before medical visitation [43]. However, *mMAP* and *GLEAM* estimates are shifted later by the delay in death reporting, likely making the dates of estimation of the methods fairly close. There is limited research on quantifying this delay, though one study found it to be 4.29 days for Mexico and 1.74 for England, with a wide range of heterogeneity between localities [44].

The uncertainty and bias of each individual method should be considered carefully. The *Divergence* methods suffer from the same challenges faced when attempting to scale CDC-measured ILI activity to the entire population [45]. In particular, scaling to case counts in a population requires estimates for $p$(visit), the probability that a person seeks medical attention for any reason, and $p$(visit | ILI) which captures health care seeking behavior given that a person is experiencing ILI; these estimates are likely to change over time, especially during the course of a pandemic. At the beginning of the pandemic, many more than usual may have paid their doctors a visit on the first sign of any ILI symptoms. Moreover, the weekly symptomatic incidence estimates from this method decrease towards the end of March, perhaps caused by a drop in health care seeking behavior after the declaration of a national emergency on March 13, 2020 and the widespread implementation of shelter-in-place mitigation strategies which may have increased the use of medical services and health providers that are not included in the ILI surveillance network, such as telehealth services and urgent care.

It is also important to note that ILI based methods are expected to be accurate only while ILI surveillance systems are operating normally (reporting tends to decrease outside of the flu season) and only while the outbreak has not yet overwhelmed hospitals and doctors. Fig 4 shows the underlying influenza surveillance data for the last five seasons. We note a sharp decrease in the total number of reported patients in late March 2020 even though the number of providers did not decrease more than is usually expected. This suggests that the ILINet signal may no longer be reliable until regular reporting patterns return. As a result, we only use ILI based methods to estimate COVID-19 symptomatic incidence early in the outbreak.

*COVID Scaling* relies on the assumption that COVID-19 positive test proportions uniformly represent the pool of all ILI patients and that shortages in testing do not bias the positive proportion. This assumption may be problematic when prior suspicion of exposure is involved, such as when health workers at a nursing home outbreak are preemptively tested, and may be a greater issue during testing shortages. In a sensitivity analysis, we computed the hypothetical impact of testing bias, finding that in the most extreme case, the true case count could be 80% of what we estimated (S3 Text).

*mMAP* is limited by assumptions of the the distribution of time from case onset to death. Furthermore, *mMAP* and *GLEAM* rely on assumptions about the IFR and asymptomatic rate; point estimates of each are uncertain, with reasonable estimates ranging from 0.65% to 1.1% for the IFR and 17% to 50% for the asymptomatic rate, which therefore yields sCFR point estimates ranging from 0.78% to 2.2% (a more detailed discussion these values is provided in the *mMAP* methods). As well, the IFR likely evolved as the pandemic progressed because treatments were improving and different subsets of the population were infected at different times. However, there is some evidence that the IFR remained stable during the beginning of the pandemic [22, 46] and a meta analysis of IFR using data until September [47] yielded similar IFR estimates to the meta-analysis using data until April [22, 48], indicating that there is not a clear decreasing trend in IFR. Both *mMAP* and *GLEAM* rely on accurate reporting of COVID-19

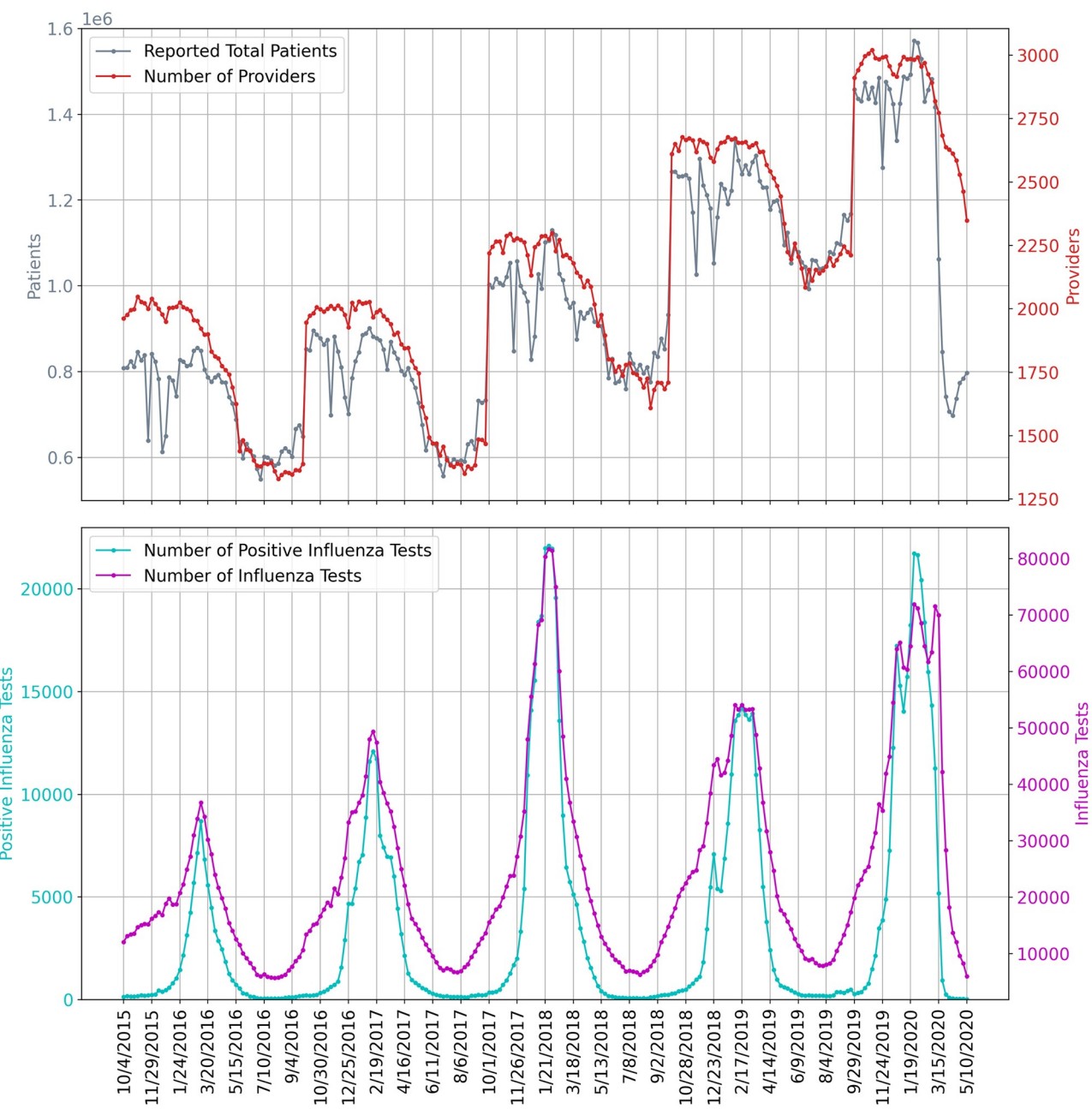

**Fig 4. The underlying influenza surveillance data for the last five seasons.** The top subplot shows the ILINet total number of patients and participating providers. The bottom subplot shows the total reported numbers of influenza tests conducted and positive influenza tests.

deaths and, in the case of adjusted *mMAP*, that excess influenza and pneumonia deaths capture all unreported COVID-19 deaths. It is likely that there are other unreported COVID-19 deaths that are not categorized as influenza and pneumonia deaths [49]. In New York City, for example, probable COVID-19 deaths (as in, not needing a test result) are being reported as COVID-19 deaths and accounted for a 42% increase in cumulative COVID-19 death counts as of April 29, 2020 [50], indicating that other locations not counting probable deaths could be missing a significant portion of deaths. Under-reporting of deaths may explain why *mMAP*

**Table 2. Comparing the four approaches to estimate COVID-19 cases in the US.**

| Approach | *Divergence* | *COVID Scaling* | *mMAP* | *GLEAM* |
|---|---|---|---|---|
| **Brief Description** | Treat COVID-19 ILI-symptomatic case count estimation as a causal inference problem. COVID's impact on ILI activity is measured using as controls a projection based on historical ILI data as well as influenza testing statistics. | Extrapolate state-level positive test percentages for COVID-19 to the weekly ILI data to estimate COVID-19 proportion in medical visits, then scale to the whole population. | Using reported COVID-19 deaths, the sCFR, and a distribution of time from cases to deaths, predict the latent case distribution. | Estimates cumulative infections using a stochastic spatially structured epidemic model, calibrated on weekly incident deaths. |
| **Data Input** | ILI activity and influenza test results. | ILI activity and COVID-19 test results. | COVID-19 deaths. | COVID-19 deaths. |
| **Model Assumptions** | 1. The divergence between predicted ILI activity for the 2019–2020 season and measured ILI activity after the start of the COVID-19 pandemic can be attributed to COVID-19.<br>2. Scaling from ILI to population is reliable. | 1. COVID-19 test reports accurately represent the pool of weekly ILI visits.<br>2. Delayed test reporting does not significantly affect positive test proportions after applying smoothing<br>3. Scaling from ILI to population is reliable. | 1. All COVID-19 deaths are reported (mMAP) or explained by excess pneumonia deaths (mMAP$^{adj}$).<br>2. The distribution of time from cases to death is log-normal.<br>3. The age-stratified IFR is the same as reported in [64] and the asymptomatic rates are 40% or 50%. | 1. Modeling estimates for the effect of school closures, smart working, and social distancing effects on the transmissibility of SARS-CoV-2.<br>2. Spatial variation of the IFR are not considered.<br>3. Differential transmissibility across age brackets is not considered.<br>4. Pre-symptomatic transmission is not modeled explicitly. |
| **Expected Bias** | Can be sensitive to model fit and changes in healthcare seeking behavior, and it will work only while ILI surveillance is reliable. | ILI visits and COVID-19 tests may capture different segments of the sick population. | May underestimate cases as many COVID-19 related deaths may go unreported or untested. | Revision to the current estimate of the IFR affects the model estimated of the total number of infections/cases. |

and *GLEAM* sometimes yield lower case estimates than *Divergence* and *COVID Scaling* even though its symptomatic case definition is more inclusive. A high-level summary of the three methods, their estimation strategy, and their assumptions are provided in Table 2.

## Conclusions

We have presented four complementary approaches for estimating the true COVID-19 cumulative (ILI-)symptomatic incidence in the United States from March 1 to May 16, 2020 at the national and state levels. The approaches rely on different datasets and modeling assumptions in order to balance the inherent biases of each individual method. While the case count estimates from these methods vary, there is general agreement among them that the actual state-level symptomatic case counts up to April 4, 2020 were likely 5 to 50 times greater than what was reported. Up to May 16, 2020, most states likely had 5 to 10 times more cases than reported, with a total estimated range of 4.9 million to 10.1 million cases over the United States.

A more accurate picture of the burden of COVID-19 is actionable knowledge that will help guide and focus public health responses. As social distancing measures are being (or have been) relaxed, some locations are experiencing a resurgence in cases. If the true case counts are near the upper bound of our estimated symptomatic case count, then a substantial proportion (up to 3% as of May 16) of the US population may have already been infected. Factoring in asymptomatic cases this could increase the proportion up to 8%. On the other hand, it is evident that the large majority of the population has not yet been exposed to COVID-19, and therefore effective, informed public health responses to future upsurges in cases will be essential in the upcoming months.

## Data and methods

### CDC ILI and virology

The CDC US Outpatient Influenza-like Illness Surveillance Network (ILINet) monitors the level of ILI circulating in the US at any given time by gathering information from physicians' reports about patients seeking medical attention for ILI symptoms. ILI is defined as having a fever (temperature of 37.8+ Celsius) and a cough or a sore throat. ILINet provides public health officials with an estimate of ILI activity in the population but has a known availability delay of 7 to 14 days. National level ILI activity is obtained by combining state-specific data weighted by state population [12]. Additionally, the CDC reports information from the WHO and the National Respiratory and Enteric Virus Surveillance System (NREVSS) on laboratory test results for influenza types A and B. The data is available from the CDC FluView dashboard [11]. We omit Florida from our analysis as ILINet data is not available for Florida.

### COVID-19 case and death counts

The US case and death counts are taken from the New York Times repository, which compiles daily reports of counts at the state and county levels across the US [51]. For the *mMAP* validation in S4 Text, the case and death counts from other countries are taken from the John's Hopkins University COVID-19 dashboard [52]. Counts are taken up until May 28, 2020.

### COVID-19 testing counts

In addition, daily time series containing positive and negative COVID-19 test results within each state were obtained from the COVID Tracking Project [53].

### US demographic data

The age-stratified, state-level population numbers are taken from 2018 estimates from the US census [54].

### Approach 1: Divergence

Viewing COVID-19 as an intervention, this approach aims to construct control time series representing the counterfactual 2019–2020 influenza season without the effect of COVID-19. While inspired by the synthetic control literature [55, 56], we are forced to construct our own controls since COVID-19 has had an effect in every state. We formulate a control as having the following two properties:

1. The control produces a reliable estimate of ILI activity, where ILI refers to the symptomatic definition of having a fever in addition to a cough or sore throat.

2. The control is not affected by the COVID-19 intervention (that is, the model of ILI conditional on any relevant predictors is independent of COVID-19).

We construct two such controls, one based on historical seasonality and one based on current virology data. We also explore a model-based method, with details in S1 Text.

**Method 1: Singular value decomposition-based historical projection.**   Unseen future ILI can be projected by fitting a time series model to historical ILI data which can account for trends that capture state-specific seasonal trends. A simple approach capable of doing this could be a simple historical weekly average of past flu seasons; however this baseline approach would lack the flexibility to incorporate the thus-far observed season-specific patterns. Instead, in our approach, we model ILI during a specific flu season, in a specific location, as the

historical average $H$ at that location plus a season-specific component $Y$:

$$ILI = H + Y$$

We produce an estimate $\hat{Y}$ of the season-specific component $Y$ by first arranging the historic ILI data into a matrix $X$ with rows corresponding to weeks in a season (time) and columns corresponding to (space) observed ILI in all spatial locations in past seasons. We then compute the most salient features of this matrix (in other words, we identify the weeks with highest variance) by factorizing the matrix $X$ using the singular value decomposition (SVD): $X = U\Sigma V^T$. The columns of $U$ form an orthonormal basis for the ILI behavior during past and fully-observed seasons. The SVD algorithm returns these columns ranked by importance by the singular values in $\Sigma$. The estimate of the season-specific component is then computed using an elastic net regression using the (first elements of the) columns of $U\Sigma$ as predictors to fit the thus-far observed ILI activity as a response variable. The unseen portion of the season is then calculated using the full length of the columns U with the regression coefficients from the elastic net fit. In other words, we make the assumption that an epidemic year can be described as a linear combination of vectors using the historic data for all locations. We scale $U$ by the singular values in $\Sigma$ so that elastic net's regularization will favor basis vectors with higher importance.

For each location, we use between 7 and 10 years of historical ILI data depending on data availability and quality. We also perform variable selection by keeping only the basis vectors $U_i$ where $\frac{\Sigma_{ii}}{trace(\Sigma)} > \frac{1}{52}$, yielding around 10 basis vectors kept depending on the year. The elastic net regularization parameters are tuned by validation on a fraction of the current season data closest to the prediction period of interest. Fig 5 shows the improvement over a historical average during the 2018–2019 season prediction period by also incorporating a season specific component to model ILI. Overall, the SVD Historical Projection performs better than the historical average baseline. The locations for which SVD Historical Projection performs worse are locations where both methods have low error.

**Method 2: Virology.**   As an alternative control, we also present an estimator of ILI activity using influenza virology results. As suggested by [19], there has been a divergence in March between CDC measured ILI activity and the fraction of ILI specimens that are

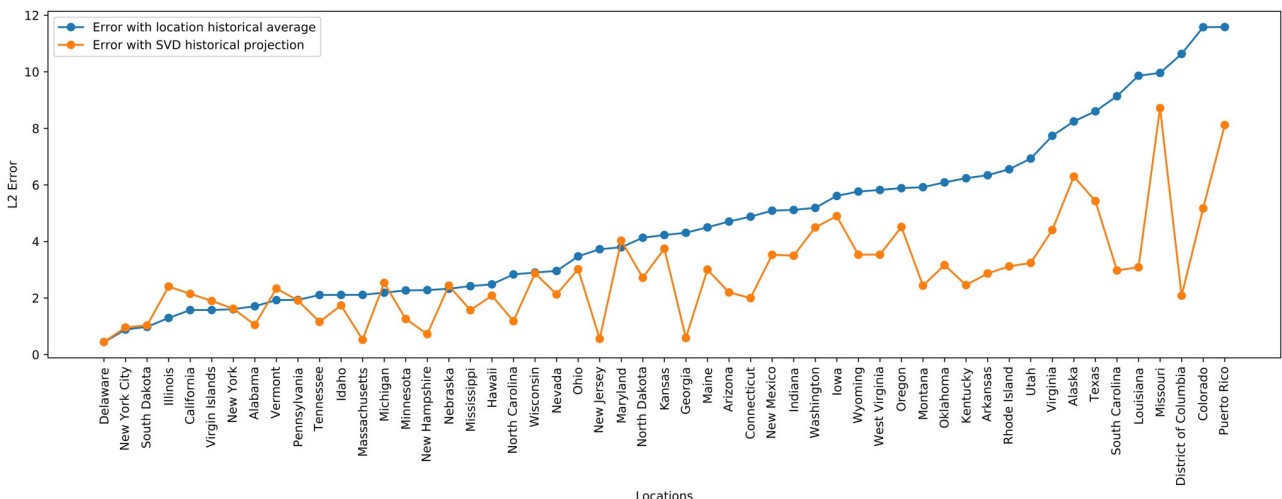

**Fig 5. L2 errors by location for March, April, and May 2019, comparing SVD Historical Projection with a baseline historical average for ILI prediction.**

influenza positive. Clinical virology time series were obtained from the CDC virologic surveillance system consisting of over 300 laboratories participating in virologic surveillance for influenza through either the US WHO Collaborating Laboratories System or NREVSS [12]. Total number of tests, total influenza positive tests, and percent positive tests are our variables of interest.

None of the three time series satisfy both properties of a valid control, as defined in, since total number of tests is directly susceptible to increase when ILI caused by COVID-19 is added. Similarly, percent positive flu tests may decrease when COVID-19 is present. On the other hand, total positive flu tests satisfies property 2, but is not a reliable indicator of ILI activity (property 1) on its own because it is highly dependent on the quantity of tests administered.

We propose a modification that satisfies the properties. Let $F_t^+$, $N_t$, $I_t$ denote positive flu tests, total specimens tested, and ILI visit counts respectively. In addition, let $F_t$ be the true underlying flu counts. For any week $t$ we assume the following relation:

$$F_t = \frac{F_t^+ \cdot I_t}{N_t}$$

There are two interpretations of this quantity: 1) It extrapolates the positive test percentage ($F_t^+/N_t$) to all ILI patients ($I_t$), a quantity known in the mechanistic modeling literature as ILI+ [57]. 2) It adjusts the number of positive tests for test frequency, which is a confounder in the relationship between the number of positive tests ($F^+$) and total flu cases ($F_t$) [58]. In S2 Text, we demonstrate over a series of examples that this estimator behaves as desired. Each estimate of $F_t$ is then scaled to population ILI cases using least squares regression over pre-COVID-19 ILI counts.

In other words, we first use virology data to estimate $F_t$ (actual flu cases causing ILI) as: percent ILI visits times percent positive for flu. Then, modeling ILI visits ($I_t$) as an affine function of $F_t$ in a normal (without COVID-19) situation, we use 2019 pre-COVID-19 data to fit the regression. This allows us to estimate the divergence after the COVID-19 intervention occurs.

**ILI case count estimation.**  To fit the above models, we estimate the ILI case count in the population from the CDC's reported percent ILI activity, which measures the fraction of medical visits that were ILI related.

In a similar fashion to the approach of [45], we can use Bayes' rule to map percent ILI activity to an estimate of the actual population-wide ILI case count. Let $p(\text{ILI})$ be the probability of any person having an influenza-like illness during a given week, $p(\text{ILI} \mid \text{visit})$ be the probability that a person seeking medical attention has an influenza-like illness, $p(\text{visit})$ be the probability that a person seeks medical attention for any reason, and $p(\text{visit} \mid \text{ILI})$ the probability that a person with an influenza-like illness seeks medical attention. Bayes' rule gives us

$$p(\text{ILI}) = \frac{p(\text{visit})}{p(\text{visit} \mid \text{ILI})} \cdot p(\text{ILI} \mid \text{visit})$$

$p(\text{ILI} \mid \text{visit})$ is the CDC's reported percent ILI activity, for $p(\text{visit})$ we use the estimate from [45] of a weekly doctor visitation rate of 7.8% of the US population, and for $p(\text{visit} \mid \text{ILI})$ we use a base estimate of 27%, consistent with the findings from [29]. Once $p(\text{ILI})$ is calculated, we multiply $p(\text{ILI})$ by the population size to get a case count estimate within the population.

**Visit propensity adjustment.**  We note that health care seeking behavior varies by region of the United States as shown in [29]. To better model these regional behavior differences, we adjust $p(\text{visit} \mid \text{ILI})$, the probability that a person with an influenza-like illness seeks medical attention, using regional baselines for the 2019–2020 influenza season [12].

Additionally, because our method estimates the increase in ILI visits due to the impact of COVID-19, we must distinguish an increase due to COVID-19 cases from an underlying increase in medical visit propensity in people with ILI symptoms. Due to the widespread alarm over the spread of COVID-19, it would not be unreasonable to expect a potential increase in ILI medical visits even in the hypothetical absence of true COVID-19 cases.

For this reason, we also explore increasing $p(\text{visit} \mid \text{ILI})$ from 27% to 35% to measure the possible effect of a change in health care seeking behavior due to COVID-19 media attention and panic. The increase of $p(\text{visit} \mid \text{ILI})$ to 35% is consistent with health care seeking behavior surveys done after the start of COVID-19 [24, 25]. The *Divergence* and *COVID Scaling* methods have *adjusted* versions which incorporate this shift as well as *unadjusted* versions that keep the baseline 27% propensity.

**Estimating COVID-19 case counts.** The ultimate goal is to estimate the true burden of COVID-19. The projection and virology predicted ILI case counts can be used to estimate CDC ILI had COVID-19 not occurred. In other words, the projection and virology predicted ILI can be used as counterfactuals when measuring the impact of COVID-19 on CDC measured ILI. The difference between the observed CDC measured ILI and the counterfactual for a given week is then the estimate of COVID-19 ILI-symptomatic case counts for that week. Fig 6 shows example observed CDC measured ILI, historical projected ILI, and virology predicted ILI. S1 Fig contains similar plots to Fig 6 for all locations. For this method as well as the following two, we start estimating COVID-19 case counts the week starting on March 1, 2020. We note that while the projection and virology ILI predictions tend to track CDC ILI well earlier in the flu season, after COVID-19 started to impact the United States there is a clear divergence between predictions and observed CDC ILI, with CDC ILI increasing while the counterfactual estimates decrease.

This method is expected to be accurate only while ILI surveillance systems are operating normally (reporting tends to decrease outside of the flu season) and only while the outbreak has not yet overwhelmed hospitals and doctors. As a result, we use ILI based methods to estimate COVID-19 symptomatic incidence only early in the outbreak, until April 4th. The disappearance of the divergence does not mean that the outbreak is over, but rather that the ILI signal is no longer reliable.

## Approach 2: COVID scaling

This approach infers the COVID-19 fraction of the total ILI by extrapolating testing results obtained from the COVID Tracking Project [53], following the same reasoning as the Virology Divergence method. That is,

$$C_t = \frac{C_t^+ \cdot I_t}{N_t^c}$$

where $C_t^+$, $N_t^c$, $I_t$ denote positive COVID-19 tests, total COVID-19 specimens, and ILI visit counts respectively.

State-level testing results were aggregated to the weekly level and positive test percentages were computed using the positive and negative counts, disregarding pending tests. Positive test counts were adjusted for potential false negatives. There are varying estimates for the false negative rate for the RT-PCR used in COVID-19 tests, with some reports suggesting rates as high as 25–30% [59, 60]. We apply a 15% false negative rate in our analysis; repeating our analysis using a range of values from 5% to 25% yielded little difference in our estimates. On the other hand, COVID-19 testing is highly specific, so we assume no false positives. Then, the number of false negatives (*FN*) can be computed from the recorded (true) positives (*TP*) and

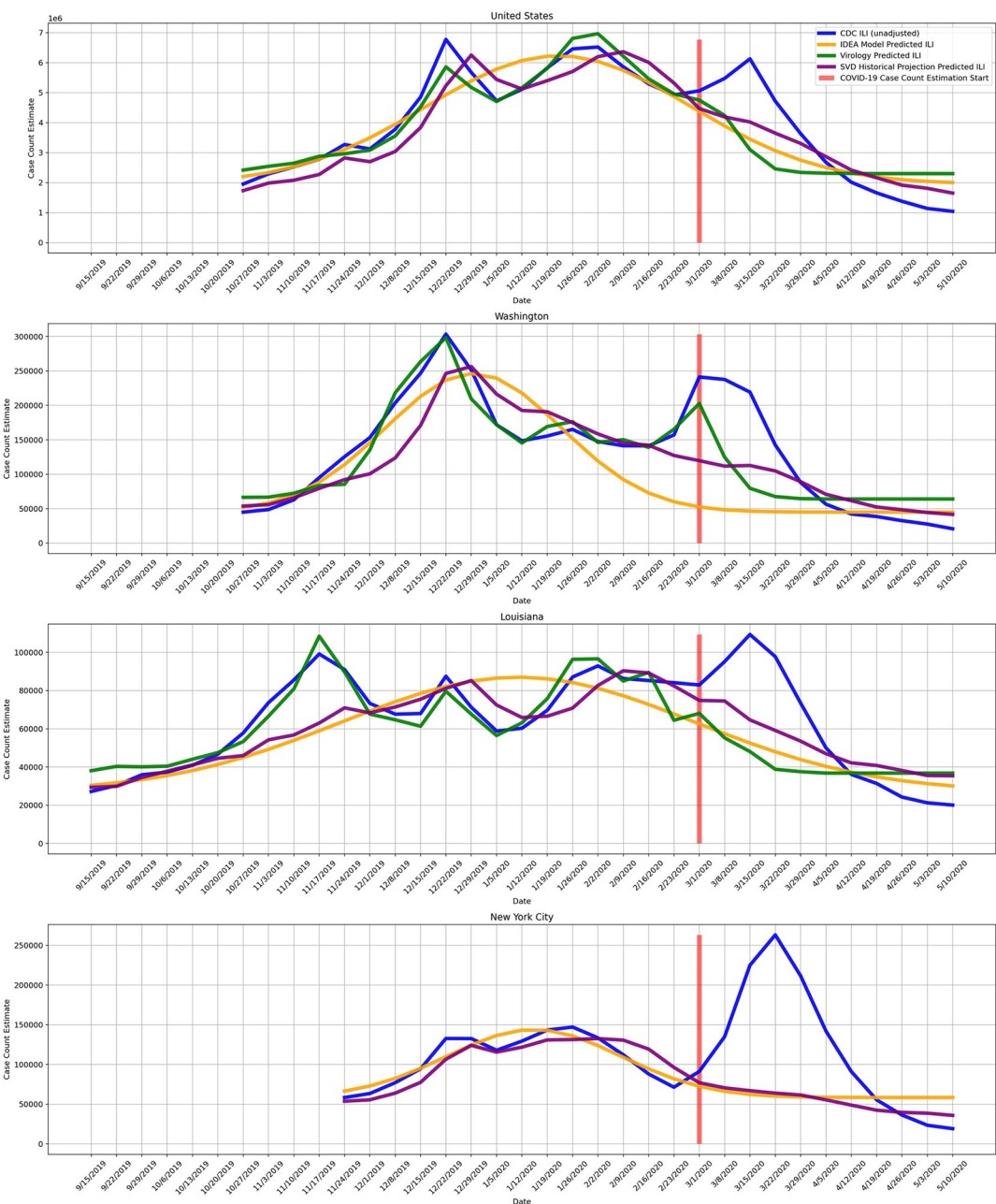

**Fig 6. COVID-19 is treated as an intervention, and we measure COVID-19 impact on observed CDC ILI, using historical projected ILI, virology predicted ILI, and historical projection predicted ILI as counterfactuals.** The difference between the higher observed CDC ILI and the lower predicted ILI is the measured impact of COVID-19. The impact directly maps to an estimate of COVID-19 ILI-symptomatic case counts. Virology predicted ILI is omitted when virology data is not available. We note that this approach is meaningful only at the beginning of the outbreak (March 2020), while ILI surveillance systems are still fully operational and before they are impacted by COVID-19. The disappearance of the divergence does not mean that the outbreak is over, but rather that the ILI signal is no longer reliable. In this figure, as a counterfactual we also include Incidence Decay and Exponential Adjustment (IDEA), a model-based method we explored with details in S1 Text.

the false negative rate (*fnr*) as

$$FN = TP \cdot \frac{fnr}{1 - fnr}$$

Because COVID-19 testing is sparse in many states, there are issues with zero or low sample sizes, as well as testing backlogs. Rather than taking the empirical positive test percentage ($C_t^+/N_t^c$), we first smoothed the test statistics over time by aggregating results over a 2-week sliding window. This has a Bayesian interpretation of combining each week's observed statistics with the prior of the previous week, weighted by relative specimen count. For convenience, $C_t^+$ and $N_t^c$ henceforth refer to these respective quantities. We also applied the same process to the ILI information to reduce noise and so that the data are comparable. This helped but did not address all issues with case backlog, so we further smoothed the COVID-19 estimates using a Bayesian spatial model:

Denote $p_{jt}$ as the prevalence of COVID-19 in a given ILI patient in state $j$ and week $t$. Assuming COVID-19 status is independent in each ILI patient, conditional on the state prevalence, the COVID-19 status of patient $i$ from state $j$ in week $t$ is

$$X_{jt}^{(i)} \sim Bernoulli(p_{jt})$$

Under the assumption that testing is applied uniformly conditional on showing ILI symptoms, the state testing results follow a Binomial distribution. We apply a spatial prior based on first-order conditional dependence:

$$p_{jt} \sim Beta(\alpha_{jt}N_{0t}, (1 - \alpha_{jt})N_{0t})$$
$$\alpha_{jt} = \frac{1}{|\mathcal{N}_j|}\sum_{k \in \mathcal{N}_j} p_{kt}$$

where $\mathcal{N}_j$ are the neighbors of state $j$. The strength of the prior was specified by setting $N_{0t}$ to be the number of total tests at the 5th quantile among all states in each week. Finally, we compute $\alpha_{jt}$ by replacing each $p_{kt}$ by their empirical estimates. Using the Beta-Binomial conjugacy we derive closed-form posterior mean estimates for $p_{jt}$:

$$\hat{p}_{jt} = \frac{C_{jt}^+ + \alpha_{jt}N_{0t}}{N_{jt}^c + N_{0t}}$$

As described previously, the weekly, state-level reported percent ILI were then multiplied by $\hat{p}_{jt}$ to get an estimate of the percent of medical visits that could be attributed to COVID-19. These values were subsequently scaled to the whole population using the same Bayes' rule method as described in *ILI Case Count Estimation* (5.1.3).

The uniform testing assumption relies on the premise that conditional on having ILI symptoms, the probability of getting tested is independent of whether the patient has COVID-19 or some other infection. This assumption is likely inexact when additional factors such as prior exposure caused biased testing towards patients more likely to be COVID-19 positive. While the impact of these factors cannot be measured in our data, we conduct a sensitivity analysis in S3 Text that models testing bias during low test availability to assess their potential impact on our estimates.

## Approach 3: Mapping mortality to COVID-19 cases

Other studies have introduced methods to infer COVID-19 cases from COVID-19 deaths using (semi-)mechanistic disease models [15] or statistical curve-fitting based on assumptions of epidemic progression [16], but, to the best of our knowledge, no methods have been proposed to directly infer COVID-19 cases without either of these assumptions.

*Mortality Map* (*mMAP*) is a time series deconvolution method that uses reported deaths to predict previous true case counts, similar to prior work on influenza [17]. *mMAP* accounts for right-censoring (i.e. COVID-19 cases that are not resolved yet) by adapting previously used methods [13]. A study of clinical cases in Wuhan found that the time in days from symptom onset to death roughly follows a log-normal distribution with mean 20.2 and standard deviation 11.6 [61]. It also found the mean time from hospitalization to death to be 13.2 days, similar to the estimate of 13.7 from a large cohort study in California [62], suggesting that the timing of disease progression is similar in the United States. Using this distribution, a smoothed time series of reported deaths, $D$ (described below), and the age-adjusted symptomatic case fatality rate (sCFR), we estimate the distribution of symptomatic cases $C$, defined at the usual time of symptom onset, using a modified expectation maximization approach. We use Bayes' rule to define the probability that there was a case on day $t$ given a death on day $\tau$.

$$p(\text{case on } t \mid \text{death on } \tau) = \frac{p(\text{death on } \tau \mid \text{case on } t) \cdot p(\text{case on } t)}{p(\text{death on } \tau)} \tag{1}$$

$D$ is the time series of reported deaths from the New York Times repository [51] ($D^{raw}$) averaged weekly. That is, $D(t) = mean[D^{raw}(max(t-3, 0)), \ldots, D^{raw}(min(t+3, t_{max})]$. The reporting of deaths depends heavily on the day of the week due to limited reporting on weekends, and we found that averaging the deaths by week significantly improves the performance of *mMAP* (more frequent convergence, smoother and more reasonable case time series).

Let $C_{d^*}$ denote the predicted distribution of when $D$ are classified as cases (i.e. are hospitalized), $C_d$ denote the predicted distribution of when $D$ and future deaths are classified as cases (so adjusted for right-censoring), and $t_{max}$ denote the most recent date with deaths reported. Let $p(\text{death on } \tau \mid \text{case on } t) = p(T = (\tau - t))$ denote the log-normal probability. *mMAP* performs the following steps:

1. Initialize the prior probability of a case on day $t$, $p_0(\text{case on } t)$, as uniform.

2. Repeat the following for each iteration $i$:

- Calculate $C_{d^*}^{(i)}$.

$$
\begin{aligned}
C_{d^*}^{(i)}(t) &= \sum_{\tau=t+1}^{t_{max}} D(\tau) \cdot p_{i-1}(\text{case on } t \mid \text{death on } \tau) \\
&= \sum_{\tau=t+1}^{t_{max}} D(\tau) \cdot \frac{p(T = (\tau - t)) \cdot p_{i-1}(\text{case on } t)}{\sum_{s=1}^{\tau-1} p(T = (\tau - s)) \cdot p_{i-1}(\text{case on } s)}
\end{aligned} \tag{2}
$$

where the denominator is equivalent to $p(\text{death on } \tau)$ in (1).

- We estimate that the proportion $p(T \leq (t_{max} - t))$ of $C_d^{(i)}(t)$ have died by $t_{max}$ and use this to adjust for right censoring.

$$C_d^{(i)}(t) = \frac{C_{d*}^{(i)}(t)}{p(T \leq (t_{max} - t))} \tag{3}$$

- Update prior probabilities

$$p_i(\text{case on } t) = \frac{C_d^{(i)}(t)}{\sum C_d^{(i)}(t)} \tag{4}$$

- Repeat until the normalized $\chi^2$ statistic descends below 1 or decreases by less than 10% on successive iterations (justification provided below):

$$\chi^2 = \frac{1}{t_{max}} \sum_{t=1}^{t_{max}} \frac{(E(t) - D(t))^2}{E(t)} \tag{5}$$

where $E(\tau) = \sum_{t<\tau} C_d^{(i)}(t) \cdot p(T = \tau - t)$ is the expected (predicted) number of deaths on day $\tau$.

3. $C_d(t)$ represents the number of cases on day $t$ that will lead to death. We scale this to estimate the number of all symptomatic cases by dividing by the sCFR.

$$C(t) = \frac{C_d(t)}{sCFR} \tag{6}$$

Interestingly, the update step for $C_d^{(i)}(t)$ in each iteration is the same as the Richardson-Lucy deconvolution step, or expectation-maximization step for the likelihood of the underlying cases, proposed for influenza [17] and for positron emission tomography (without right-censoring) [63], albeit with different notation in each study. S4.1 Text demonstrates this equivalence and discusses the mathematical justification this provides for *mMAP*. The influenza paper demonstrates that under the true parameters (or true case time series), $D(t)$ would follow a Poisson distribution with mean $E(t)$ and therefore the chi-squared statistic (Eq (5)) would have expectation 1. Thus it is useful to iterate until (5) is less than one for the first time and stopping there to avoid over-fitting the noise in the observed death data [17]. We also stopped if (5) changed by less than 10% for successive iterations because for locations with large enough death numbers (United States, New York, New Jersey, and Texas) the value of (5) never descended below 1.

S4.2 Text demonstrates that *mMAP* successfully predicts cases using simulated and reported deaths from six countries, providing further justification for this method.

If one were interested in estimating the incidence of all cases—symptomatic and asymptomatic—$C_d(t)$ would need to be divided by the infection fatality ratio (IFR) in step 3 (Eq 6). For the sake of comparison with the ILI-based methods in this study, we chose to use sCFR in the denominator in Eq 6 to estimate the incidence of just the symptomatic cases. The national sCFR values used are 2.2% and 1.1% for the unadjusted and adjusted method. These values were found by adjusting the IFR estimates (1.1% and 0.65%) with an assumed 50% and 40% asymptomatic rate, respectively (estimates of the percentage of asymptomatic cases range from 17/18% [6, 26] to 50% [27, 28] and the CDC puts 40% as the best point estimate of this number [8, 23]). The first IFR value comes from an analysis of individual case data in China and repatriated Chinese citizens in January and February to estimate the fatality ratio for all—symptomatic and asymptomatic—infections [21]. The second value comes from a meta-analysis of

published IFR values and is the CDC best point estimate of the IFR [22, 48]. The sCFR estimates for each state are adjusted using the age-stratified fatality rate [64] and the population age structure provided by the US census [54], with values ranging from 1.3% in Utah to 2.3% in Florida.

**Accounting for unreported COVID-19 deaths.**   While *mMAP* assumes all COVID-19 deaths are reported, some deaths will be unreported because of limited testing and false negative results [65, 66]. Previous research on the H1N1 epidemic estimated that the ratio of lab-confirmed deaths to actual deaths caused by the disease was 1:7 nationally [67] and 1:15 globally [68]. While the actual rate of under-reporting is unknown, we include an adjustment, mMAP$^{adj}$, that attributes excess influenza and pneumonia deaths to COVID-19, as has been done in previous studies [65].

The CDC reports weekly reported influenza and pneumonia deaths and expected influenza and pneumonia deaths based on historical trends for each state [69]. We estimate that the number of un-classified COVID-19 deaths for a given location each week, $D_U(w)$, is $max(0,$ reported deaths—expected deaths). Nationally, this leads to two peaks of $D_U(w)$—the first, larger peak in March in April accounting for 2,791 deaths and a smaller peak in July and August accounting for 221 deaths—with values of zero excess deaths almost every other week. The daily deaths used for mMAP$^{adj}$, $D^{adj}(t)$, are the sum of the reported deaths, $D(t)$, and the average of the weekly excess deaths.

$$D^{adj}(t) = D(t) + \frac{1}{7}D_U(w), \text{ where } t \in w \tag{7}$$

## Approach 4: Global epidemic and mobility model

The Global Epidemic and Mobility model is an individual-based, stochastic, and spatial epidemic model. *GLEAM* uses real-world data to perform *in silico* simulations of the spatial spread of infectious diseases at the global level. In the model, the world is divided into over 3,200 geographic subpopulations constructed using a Voronoi tessellation of the Earth's surface. Subpopulations are centered around major transportation hubs and integrate data on the population such as age specific contact patterns [70], short-range (i.e. commuting) and long-range (i.e. flights) mobility data from the Offices of Statistics for 30 countries on 5 continents as well as the Official Aviation Guide (OAG) and IATA databases (updated in 2019) [71, 72]. The model has been used extensively to analyze previous epidemic such as the H1N1 2009 pandemic and the Zika epidemic in the Americas [73–75], and to simulate the early spreading of COVID-19 in mainland China [18].

We use the model to analyze the spatiotemporal spread and magnitude of the COVID-19 epidemic in the continental US. For COVID-19 the model adopts a classic *SLIR* disease characterization in which individuals can be classified into four compartments: susceptible, latent, infectious, or removed. Susceptible individuals become latent through interactions with infectious individuals. During both the latent and infectious stages we assume that individuals are able to travel. Following the infectious period, individuals then progress into the removed compartment where they are no longer able to infect others, meaning they have either recovered, been hospitalized, isolated, or have died. The disease dynamic does not explicitly describe the pre-infectious period that is implicitly accounted for in the infectious stage and the length of the generation time. *GLEAM* is able to simulate explicitly the disease dynamic at the individual level.

Approximate Bayesian Computation is used to estimate the posterior distribution of the basic parameters of the model. The prior distribution of the parameters and the calibration of

the global model for COVID-19 is reported in [18]. Within the US, we have implemented domestic airline traffic reductions and local commuting pattern reductions. The magnitude of these reductions is based on the analysis of data from millions of (anonymized, aggregated, privacy-enhanced) devices [76] and official airline data from OAG. We consider two major social distancing periods in the US. The first period includes mitigation policies widely adopted on March 16, 2020 [20], including system-wide school closures, work from home policies (smart work), and reduction in casual social interactions in the community. The second period refers to the issuing in more than 41 states of "stay at home" or "shelter in place" orders starting on April 1, 2020. The impact of these mitigation policies is reflected in specific contact patterns calculated in the model's synthetic populations on the different layers where individuals interact: households, schools, workplaces, and in the general community. We also consider in each state the progression into reopening phases after April 30th, 2020.

As our model considers contact matrices for different settings, namely households, schools, workplaces and community contacts [70, 77], we quantify the decrease in contacts that individuals have in each of these environments. To implement school closures in the United States we follow [78] where authors study the effects of school closure in the context of seasonal influenza epidemics. According to the date when schools were closed in the different states we consider a reduction of contacts in all individuals attending an educational institution [79, 80]. This intervention was applied at state level.

Following the school closure, most US states issued a "stay at home order". In this case, we consider that only contacts in the household and essential workplaces were available. Using the COVID-19 Community Mobility reports [81] we compute the relative reduction on the number of contacts in workplaces, and community interaction as well as the relative reduction in the intra-country mobility. From the Google mobility reports we use the field `work-places percent change from baseline` to infer contacts reduction in workplaces, the average of the fields `retail and recreation percent change from base-line` and `transit stations percent change from baseline` for the general community settings. The Google mobility report provides the percentage change $r_l(t)$ on day $t$ of total visitors to specific locations $s$ with respect to a pre-pandemic baseline. We turn this quantity into a rescaling factor for contacts such as $\omega_s(t) = \omega_s(1 + r_l(t)/100)^2$, by considering that the number of potential contacts per location scales as the square of the number of visitors.

When the interventions are relaxed the mobility reduction is relaxed accordingly. Finally we explore different level of overall transmissibility reduction (0–30%, step 10%) due to the awareness of population and behavioral changes starting at the date of the state of the emergency in the US.

By using the global calibration we generate an ensemble of epidemic models defined by the posterior distribution of the parameters and the interventions in each state that provides the weekly number number of new deaths by using available estimates of infection fatality rate [21, 23, 82]. For each model that satisfies the global calibration, we use the Akaike Information Criterion (AIC) with information loss $\Delta_i < 9$. The selected models define the median and 95% CI for cumulative infections in each state (Fig 7). The estimated total number of infections can be adjusted to provide an estimate of COVID-19 symptomatic cases by reducing the predictions by an estimated asymptomatic rate of 40% [22, 23]. In Fig 7, we report the model estimates of the cumulative number of infections on May 16, 2020 compared to the number of cases reported through that date within each state. We see a strong correlation between the reported cases and our model's estimated number of infections, (Pearson's correlation coefficient on log-values 0.98, $p < 0.001$). If we assume that the number of reported cases and simulated infections are related through a simple binomial stochastic sampling process, we find

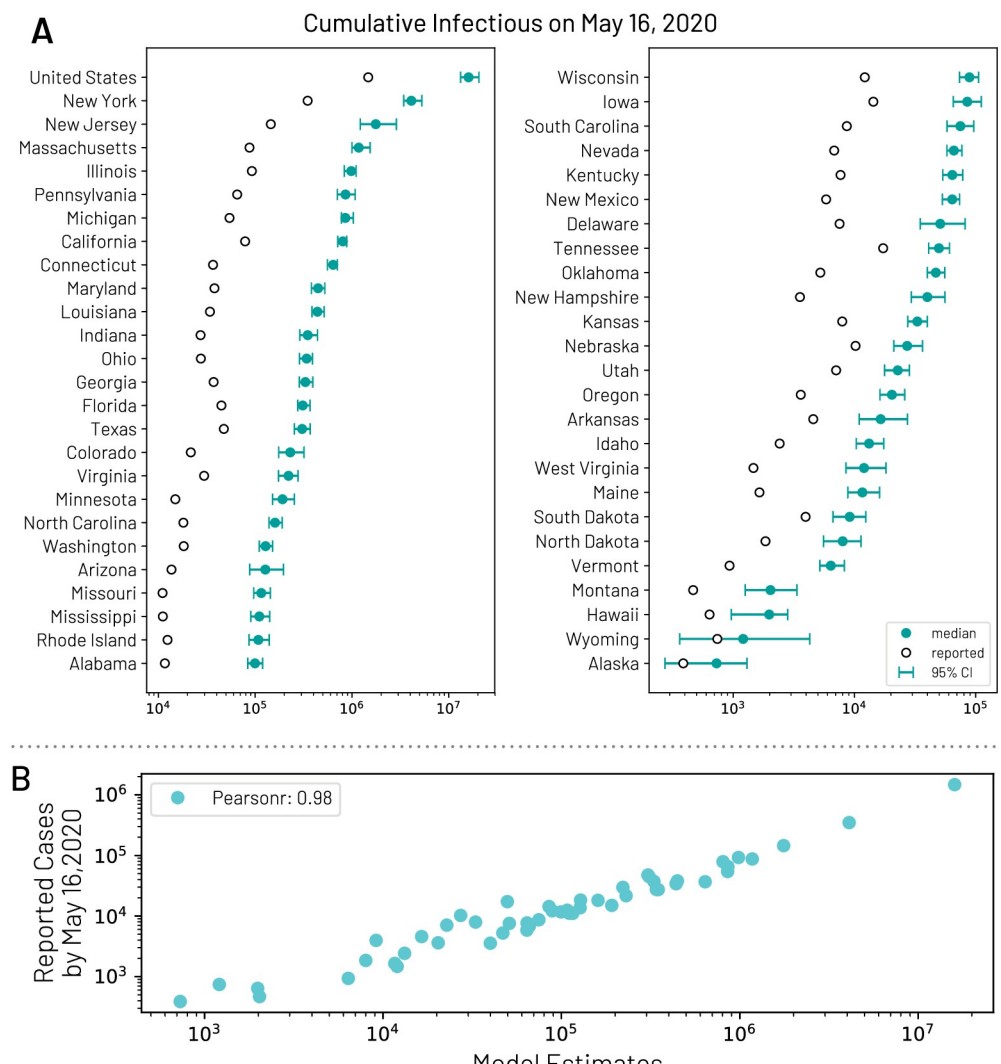

**Fig 7.** (A) Model estimates of the cumulative number of infections using the *GLEAM* model by May 16, 2020 for each state. (B) Correlation between the number of reported cases of COVID-19 for each state and the model estimates of the total number of infections by May 16, 2020.

that the median ascertainment rate of detecting an infected individual by May 16, 2020 is 11.2% (95%CI: [6.4%, 40.5%]). The detailed model's results are publicly available at https://covid19.gleamproject.org/.

## Aggregation of estimates

The divergence-based methods predict national COVID-19 symptomatic incidence directly using national ILI data. *mMAP* and *GLEAM* predict national symptomatic incidence using national death data, while *COVID Scaling* estimates national symptomatic incidence by aggregating the case estimates from each state.

The *Divergence* and *COVID Scaling* methods provide separate case estimates for each week within the studied period, which are summed to the total cumulative case estimates. *mMAP* and *GLEAM* provide daily estimates which are further aggregated by week.

## Supporting information

**S1 Fig. Divergence by location.** Figures A and B in S1 Fig show the *Divergence* approach model fits for all available locations. COVID-19 is treated as an intervention, and we measure COVID-19 impact on observed CDC ILI, using IDEA model predicted ILI, virology predicted ILI, and historical projection predicted ILI as counterfactuals. The difference between the higher observed CDC ILI and the lower predicted ILI is the measured impact of COVID-19. The impact directly maps to an estimate of COVID-19 ILI-symptomatic case counts. Virology-predicted ILI is omitted when virology data is not available. We note that model fit quality varies by location. CDC reported ILI activity is plotted in blue, historical projection predicted ILI is plotted in purple, IDEA model predicted ILI is plotted in orange, and virology predicted ILI is plotted in green. We note that this approach is meaningful only at the beginning of the outbreak (March 2020), while ILI surveillance systems are still fully operational and before they are impacted by COVID-19. The disappearance of the divergence does not mean that the outbreak is over, but rather that the ILI signal is no longer reliable. As a reference, Figures C and D in S1 Fig show the model fits for the same locations during the COVID-free 2018–2019 flu season.
(PDF)

**S2 Fig. Time series plots for all methods.** Figures A and B in S2 Fig show the cumulative estimated counts for each week over the entire study period of March 1, 2020 to May 16, 2020, compared with cumulative reported counts, in each location in the United States. The solid and dotted lines indicate adjusted and unadjusted methods, respectively. Due to the seasonal nature of ILI information, estimates from all approaches besides *mMAP* and *GLEAM* are limited to April 4, 2020.
(PDF)

**S1 Text. A third divergence method: Incidence decay and exponential adjustment model.** We explore an additional model-based method for ILI counterfactual estimation for the *Divergence* approach.
(PDF)

**S2 Text. Virology-based estimation.** Theoretical backing for virology-based estimation.
(PDF)

**S3 Text. COVID scaling sensitivity analysis.** Sensitivity analysis on different assumptions of *COVID Scaling*.
(PDF)

**S4 Text. Mortality-MAP analysis.** Theoretical backing for Mortality-MAP method.
(PDF)

## Acknowledgments

Gonzalo Mena for contributions to optimize the code.

## Author Contributions

**Conceptualization:** Fred S. Lu, Andre T. Nguyen, Nicholas B. Link, Alessandro Vespignani, Marc Lipsitch, Mauricio Santillana.

**Data curation:** Fred S. Lu, Andre T. Nguyen, Nicholas B. Link, Mathieu Molina, Jessica T. Davis, Matteo Chinazzi, Xinyue Xiong.

**Formal analysis:** Fred S. Lu, Andre T. Nguyen, Nicholas B. Link, Mauricio Santillana.

**Funding acquisition:** Mauricio Santillana.

**Investigation:** Fred S. Lu, Andre T. Nguyen, Nicholas B. Link, Jessica T. Davis, Matteo Chinazzi, Xinyue Xiong, Alessandro Vespignani, Marc Lipsitch, Mauricio Santillana.

**Methodology:** Fred S. Lu, Andre T. Nguyen, Nicholas B. Link, Mathieu Molina, Jessica T. Davis, Matteo Chinazzi, Xinyue Xiong, Alessandro Vespignani, Marc Lipsitch, Mauricio Santillana.

**Project administration:** Mauricio Santillana.

**Resources:** Mauricio Santillana.

**Supervision:** Mauricio Santillana.

**Validation:** Fred S. Lu, Andre T. Nguyen, Nicholas B. Link, Mauricio Santillana.

**Visualization:** Fred S. Lu, Andre T. Nguyen, Nicholas B. Link, Mathieu Molina, Jessica T. Davis, Matteo Chinazzi, Xinyue Xiong, Mauricio Santillana.

**Writing – original draft:** Fred S. Lu, Andre T. Nguyen, Nicholas B. Link, Mauricio Santillana.

**Writing – review & editing:** Fred S. Lu, Andre T. Nguyen, Nicholas B. Link, Mathieu Molina, Jessica T. Davis, Matteo Chinazzi, Xinyue Xiong, Alessandro Vespignani, Marc Lipsitch, Mauricio Santillana.

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
