## [Decision Letter · Decision Letter 0]

12 Oct 2020

Dear Dr Santillana,

Thank you very much for submitting your manuscript "Estimating the Early Outbreak Cumulative Incidence of COVID-19 in the United States: Three Complementary Approaches" for consideration at PLOS Computational Biology.

As with all papers reviewed by the journal, your manuscript was reviewed by members of the editorial board and by several independent reviewers. In light of the reviews (below this email), we would like to invite the resubmission of a significantly-revised version that takes into account the reviewers' comments. It would be useful also if you could update your estimates.

We cannot make any decision about publication until we have seen the revised manuscript and your response to the reviewers' comments. Your revised manuscript may be sent to reviewers for further evaluation.

Sincerely,

Cecile Viboud

Associate Editor

PLOS Computational Biology

Nina Fefferman

Deputy Editor

PLOS Computational Biology

Reviewer's Responses to Questions

**Comments to the Authors:**

Reviewer #1: I appreciate the opportunity to review “Estimating the Early Outbreak Cumulative Incidence of COVID-19 in the United States: Three Complementary Approaches” by Lu et al. The authors utilize influenza-like illness surveillance data, COVID testing volume, and reported COVID deaths to estimate cumulative symptomatic COVID cases in US states. Though I think this work is important and that the manuscript well-written, I have a few concerns regarding the authors’ methodology. Please see my comments below.

Results

Line 64: I recommend explicitly mentioning that the “Divergence” and “COVID Scaling” approaches estimate symptomatic COVID cases up to the week of April 4th and that the “mMaP” approach estimates cases up to May 16th.

Line 95: It’s briefly touched on in the discussion but I think it’s important to acknowledge that there was a “worried well” spike in ILI visits at general practitioner offices at the beginning of the epidemic in the US (early March) and then a sharp decline in ILI due to stay-at-home orders, increased use of tele-health services, seeking healthcare at providers that are not included in ILI surveillance (e.g., urgent care, emergency rooms), and the end of the flu season.

Discussion

Line 232: ILI surveillance does not likely accurately capture ILI dynamics for elderly individuals residing in nursing homes (the individuals most at risk for severe COVID and death).

Methods

General comment: I recommend including uncertainty estimates for ILI projections, which would in turn enable uncertainty estimates for COVID burden for the approaches that utilize ILI data.

Line 324: The IDEA model/Farr’s Law seems too simplistic for projecting ILI cases in the absence of COVID. Did the authors assess whether this method accurately captures ILI trajectories in prior seasons? It seems more appropriate to use a time series model fit to historical ILI data from several past seasons (state-level data are available going back to 2010-2011), which can account for seasonality in each state and provide prediction intervals for ILI projections.

Line 369: I may have misunderstood but it seems that the authors multiplied % flu positive in 2020 by ILI counts from 2019. If this is the case, I recommend utilizing ILI data for 2010/2011 to 2018/2019 to compute an ILI “seasonal baseline” with 95% confidence intervals for each state, rather than applying 2019 ILI counts to 2020 dates.

Line 392: Did the authors also explore how the decline in health-seeking behavior after stay-at-home measures potentially impacted p(visit|ILI)?

Figures: Please define "unadjusted" and "adjusted" in figure captions.

Reviewer #2: The authors present an analysis of COVID testing and ILI data to infer the true number of symptomatic COVID-19 cases in the US. This is of significant interest to researchers and policymakers. Additionally, using existing surveillance systems (such as ILI surveillance) is an important methodological approach that has been underused. However, I would recommend some revisions and updating the manuscript to reflect recent literature that estimates true infection rates, such as [1].

Major points:

1) Is the assumption of constant sCFR and IFR through time justified? I realize that the literature on IFR is constantly evolving, but increasing quality of care as well as changes in who is getting infected (e.g., nursing home outbreaks early in the pandemic) might have shifted the overall IFR (and thus sCFR) over time. Also, does adding uncertainty in the asymptomatic fraction substantially change the results? Given that uncertainty in IFR is incorporated, it seems odd to use a point estimate for the asymptomatic fraction, given that it has a substantial range associated with it (and, again, could be changing over time as the infected population changes). In any case, if these sources of uncertainty cannot be incorporated due to lack of data, I think it is important to at least address them in the discussion.

2) Why only focus on excess pneumonia deaths to adjust for underreporting of COVID-19 deaths? I believe that the CDC provides state-level data on all-cause mortality compared to COVID-19 mortality (https://www.cdc.gov/nchs/nvss/vsrr/covid19/excess_deaths.htm#data-tables), and these estimates could be used to provide an upper bound on true deaths for the whole dataset.

3) Is the COVID scaling approach fair? Even after accounting for sparse case counts, testing backlogs, and false negatives, I find the assumption of uniformly applied testing problematic (after line 431—line numbering seems to have gone awry in this section). Testing shortages, especially early in the epidemic, may have biased testing toward higher-risk subjects. As testing capacity increased across states, test positivity may have gone down simply due to testing more people, while true prevalence (and therefore, presumably fraction of ILI infected with COVID) remained relatively constant. While this limitation is mentioned in lines 254-255, I think that additional discussion is needed, as this assumption of unbiased testing is almost certainly broken.

Minor points:

1) I think it would help with clarity to provide the names for the three approaches (divergence, scaling, and mortality) in the introduction.

2) Line 282: “(up to 4%) of the US population may have already been infected.” It would be good to specify the date here, since I assume that this was using the estimates of true infections at the end of May. In that case, the following statement that “subsequent waves of infection may decrease in magnitude” should be revaluated in the context of extant case data for the second wave of infection that occurred during the summer.

3) Line 353: I think a reference to either the supplement or previous section is missing.

4) It seems that the mortality method assumes that deaths are reported on the actual date of death. It is not clear that all states report deaths in this way, and I am curious if accounting for delays in death reporting could be incorporated into the MAP framework.

5) References 70 and 71 seem to be misformatted.

6) Supplement, after equation 7 should “P(t=0)” be “p(t=0)?” I didn’t quite follow what the change in summation limits here was.

[1] Wu, S.L., Mertens, A.N., Crider, Y.S. et al. Substantial underestimation of SARS-CoV-2 infection in the United States. Nat Commun 11, 4507 (2020). https://doi.org/10.1038/s41467-020-18272-4

**Have all data underlying the figures and results presented in the manuscript been provided?**

Reviewer #1: **No: **The authors state that data will be made available upon acceptance.

Reviewer #2: **No: **Authors have said that data will be provided in a repository upon acceptance. It is not currently available.

PLOS authors have the option to publish the peer review history of their article (what does this mean?). If published, this will include your full peer review and any attached files.

Reviewer #1: No

Reviewer #2: No
---

## [Decision Letter · Decision Letter 1]

11 Mar 2021

Dear Dr Santillana,

Thank you very much for submitting your revised manuscript "Estimating the cumulative incidence of COVID-19 in the United States using influenza surveillance, virologic testing, and mortality data: four complementary approaches" for consideration at PLOS Computational Biology. The paper went back to the two reviewers, who are very pleased with the revisions. One of the reviewers has made a few additional light suggestions. Based on the reviews, we are likely to accept this manuscript for publication, providing that you modify the manuscript according to the review recommendations.

Sincerely,

Cecile Viboud

Associate Editor

PLOS Computational Biology

Nina Fefferman

Deputy Editor

PLOS Computational Biology

[LINK]

Reviewer's Responses to Questions

**Comments to the Authors:**

Reviewer #1: I appreciate the authors' careful attention in addressing my comments on the previous submission. I do not have any additional feedback for the revised manuscript.

Reviewer #2: I thank the authors for their thoughtful responses to my comments and concerns and I believe that the addition of GLEAM to the methods tested is helpful. However, I think would be helpful to provide more information on the particulars of the authors' implementation of GLEAM. Specifically:

1) Line 619: "We assume varying levels of effectiveness of the mitigation policies...":

It was unclear to me how this was accomplished. Does this mean that effectiveness varied based on which policies different states adopted, or did the same policy adopted in two different states have different effectiveness?

2) Line 621: "We then perform model selection...": This is related to my previous point, but I think more clarity is needed regarding what precise models were compared.

3) Are the specific assumptions about the prior distribution of key parameters for the U.S. model available (e.g., incubation period, IFR, duration of infectiousness, etc.)? Are they the same as the assumptions made for the global calibration of the model [1]?

4) Does the model's assumption of no pre-symptomatic transmission affect the estimates of the true number of infections? This might be useful to add to Table 2, or as a point in the discussion of limitations.

[1] Chinazzi M, Davis JT, Ajelli M, Gioannini C, Litvinova M, Merler S, Pastore Y Piontti A, Mu K, Rossi L, Sun K, Viboud C, Xiong X, Yu H, Halloran ME, Longini IM Jr, Vespignani A. The effect of travel restrictions on the spread of the 2019 novel coronavirus (COVID-19) outbreak. Science. 2020 Apr 24;368(6489):395-400. doi: 10.1126/science.aba9757. Epub 2020 Mar 6. PMID: 32144116; PMCID: PMC7164386.

**Have all data underlying the figures and results presented in the manuscript been provided?**

Reviewer #1: **No: **The authors state "The data will be held in a public repository after acceptance at " ext-link-type="uri" xlink:type="simple">https://github.com/andrenguyen/mil-covid19-usa". The github repo is not currently accessible.

Reviewer #2: Yes

PLOS authors have the option to publish the peer review history of their article (what does this mean?). If published, this will include your full peer review and any attached files.

Reviewer #1: No

Reviewer #2: No

Figure Files:

Data Requirements:

Reproducibility:

References:

---

## [Editor Report · Decision Letter 2]

18 Apr 2021

Dear Dr. Santillana,

Thank you very much for submitting your revised manuscript "Estimating the cumulative incidence of COVID-19 in the United States using influenza surveillance, virologic testing, and mortality data: four complementary approaches" for consideration at PLOS Computational Biology.

The revised manuscript went back to the reviewers, who agree that their major comments have been addressed. Reviewer 2 has made a few residual comments/suggestions, which we expect you will be able to address easily. We are likely to accept this manuscript for publication, providing that you modify the manuscript according to the review recommendations.

Sincerely,

Cecile Viboud

Associate Editor

PLOS Computational Biology

Nina Fefferman

Deputy Editor

PLOS Computational Biology

[LINK]

Figure Files:

Data Requirements:

Reproducibility:

To enhance the reproducibility of your results, we recommend that you deposit your laboratory protocols in protocols.io, where a protocol can be assigned its own identifier (DOI) such that it can be cited independently in the future. Additionally, PLOS ONE offers an option to publish peer-reviewed clinical study protocols. Read more information on sharing protocols at https://plos.org/protocols?utm_medium=editorial-emailutm_source=authorlettersutm_campaign=protocols

References:

---

## [Editor Report · Decision Letter 3]

22 Apr 2021

Dear Dr. Santillana,

We are pleased to inform you that your manuscript 'Estimating the cumulative incidence of COVID-19 in the United States using influenza surveillance, virologic testing, and mortality data: four complementary approaches' has been provisionally accepted for publication in PLOS Computational Biology.

Best regards,

Nina H. Fefferman

Deputy Editor

PLOS Computational Biology

Nina Fefferman

Deputy Editor

PLOS Computational Biology

---

## [Editor Report · Acceptance letter]

4 Jun 2021

PCOMPBIOL-D-20-01068R3 

Estimating the cumulative incidence of COVID-19 in the United States using influenza surveillance, virologic testing, and mortality data: four complementary approaches

Dear Dr Santillana,

I am pleased to inform you that your manuscript has been formally accepted for publication in PLOS Computational Biology. Your manuscript is now with our production department and you will be notified of the publication date in due course.

With kind regards,

Olena Szabo
